# Hierarchical Policy Learning via Spectral Decomposition

Shuxin Cao [1]  Liquan Wang [1]  Walker Byrnes [1 2]  Yiye Chen [1]  Yilun Du [3]  Animesh Garg [1]

## Abstract

In this paper, we identify a semantic decomposition in robot action sequences, separating task-level motion intent from execution-level refinements. By analyzing actions in the spectral domain using the discrete cosine transform (DCT), we observe that low-frequency components capture global motion trajectories, while high-frequency components encode precise timing, alignment, and contact behaviors. Motivated by this structure, we propose Causal Spectral Policy (**CSP**), which models action generation as a causal coarse-to-fine process: coarse motion is predicted from observation and language, and fine corrections are generated conditionally on the realized trajectory. Across simulation and real-world evaluations, **CSP** consistently outperforms strong baselines on precision-sensitive manipulation tasks. Additionally, we propose human-inspired teleoperation noise injection as a data augmentation method under which our approach demonstrates strong robustness to noisy demonstrations.

https://causal-spectral-policies.github.io/.

## 1. Introduction

Robot manipulation is a combination of long-horizon trajectory planning and fine-grained reactive control. On one hand, manipulation behaviors depend on long-horizon structure, such as the coarse trajectory of an end-effector. On the other hand, fine-grained and reactive control governs the precision and timing of the action. Prior works typically represent actions directly in the time domain and predict actions autoregressively or as fixed-length chunks while treating all temporal components as equally important. (Chi et al., 2023; Haldar et al., 2024; Zhao et al., 2023) As a

[1]Georgia Institute of Technology, Atlanta, Georgia, USA [2]Georgia Tech Research Institute, Atlanta, Georgia, USA [3]Harvard University, Cambridge, Massachusetts, USA. Correspondence to: Shuxin Cao <shuxin.cao@gatech.edu>.

*Proceedings of the 43$^{rd}$ International Conference on Machine Learning*, Seoul, South Korea. PMLR 306, 2026. Copyright 2026 by the author(s).

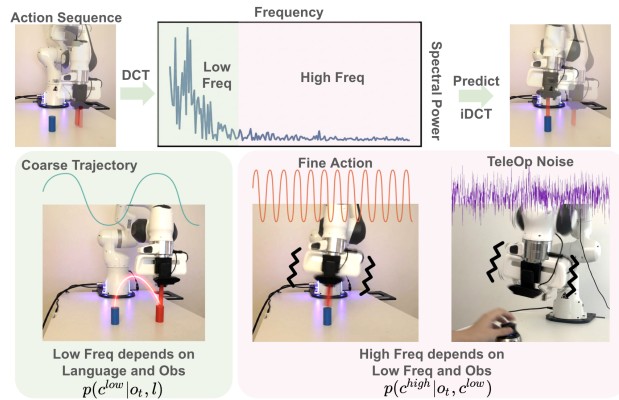

*Figure 1.* Action sequences admit a coarse-to-fine structure in the spectral domain. **CSP** predicts low-frequency motion from observation and language, then generates high-frequency corrections conditioned on the coarse trajectory.

result, these methods struggle to explicitly capture the hierarchical temporal structure in robot actions, particularly in speed and precision-sensitive manipulation tasks.

To confirm our intuition, we convert robot action sequences into the spectral domain using the Discrete Cosine Transform (DCT). We then selectively remove high-frequency components, reconstruct the resulting trajectories, and perform roll-out on real robots to examine their behaviors. Extensive real-world and simulation experiments show that removing high-frequency components largely preserves the coarse structure of motion and is often sufficient for coarse task execution, but systematically degrades fine-grained accuracy leading to errors in timing, alignment, and contact that accumulate during execution and cause failure in precision-critical stages. Real-robot demonstrations collected through teleoperation further amplify this issue, as latency, bandwidth constraints, and execution variability disproportionately affect short-timescale corrections.

Motivated by these observations, we hypothesize robot action learning is most effective with an inherent coarse-to-fine temporal structure. We propose a novel *Causal Spectral Policy* (**CSP**) to explicitly decompose robot actions in the spectral domain into low-frequency (coarse) and high-frequency (fine) components and model them separately, allowing policies to capture stable task-level behavior before predicting fine-grained corrections conditioned on the coarse plan.

We evaluate this hypothesis using simulated manipulation benchmarks, noise-perturbed demonstrations, and real-robot experiments on precision manipulation tasks involving temporal jitter and corrective motion. Our results demonstrate that an explicit coarse-to-fine action structure improves robustness and execution accuracy under realistic conditions.

Our contributions are threefold: (1) we identify a fundamental asymmetry in robot action learning, showing that coarse and fine action components play distinct causal roles; (2) through controlled frequency-domain interventions, we demonstrate that high-frequency actions are essential for precision despite contributing little to overall motion energy; and (3) we propose and validate a spectral coarse-to-fine policy learning framework that improves reliability on precision- and speed-sensitive tasks in both simulation and real-robot settings.

## 2. Related Work

### 2.1. Action Representation and Temporal Structure in Robot Learning

Finding effective action representations remains an open problem in robot learning, with diverse architectures proposed for action prediction. Most existing methods operate in the time domain: autoregressive policies generate discrete action tokens sequentially (Zhao et al., 2023; Haldar et al., 2024; Black et al., 2024), while diffusion- and flow-based approaches predict continuous fixed-length action chunks (Chi et al., 2023; Hou et al., 2025; Intelligence et al., 2025; Jiang et al., 2025; Wang et al., 2024). Although successful, both paradigms treat all timesteps uniformly and are sensitive to temporal resolution. Recently, frequency-domain representations have been explored to improve efficiency and robustness, with strong empirical benefits demonstrated in image generation (Yu et al., 2025; Ning et al., 2025; Liu & Tang, 2025; Li et al., 2024) and audio processing (Kong et al., 2021; Wang et al., 2025). In robotics, FAST (Pertsch et al., 2025) compresses action sequences using discrete cosine transforms (DCT) to reduce redundancy, while FreqPolicy (Yu et al., 2025) extends this representation with diffusion-based decoding. Existing frequency-based methods primarily emphasize representation efficiency and do not explicitly model the causal relationship between coarse motion planning and fine-grained execution, which is the focus of our work.

### 2.2. Frequency-Based and Coarse-to-Fine Action Modeling

To improve controllability in action generation, prior work has explored coarse-to-fine decomposition strategies (Johns, 2021; Cui et al., 2025), separating global motion from fine manipulation. Most approaches implement this through progressive refinement over temporal resolution, applied to both autoregressive policies (Gong et al., 2025) and flow-based models (Su et al., 2025b;c; Oh et al., 2025; Su et al., 2025a). FreqPolicy (Yu et al., 2025) extends this idea to the frequency domain by iteratively unmasking spectral coefficients from low to high frequency.

### 2.3. Robust Imitation Learning with Noisy Demonstrations

Nearly all imitation learning methods require high quality human demonstrations. However, studies have shown the quality of human demonstration data has a considerable impact on model performance and is not well controlled between datasets (Belkhale et al., 2023; Sakr et al., 2025; 2022). Work in the field of neuroscience has revealed common structures in human movements, possessing temporally correlated corrective movements (Faisal et al., 2008; Van Beers et al., 2004; Todorov & Jordan, 2002b) which scale along with the distance of movement (Todorov & Jordan, 2002a; Harris & Wolpert, 1998b). Prior work has sought to model and filter out noise from demonstrations prior to learning (Chen et al., 2026; Huang et al., 2024) or reformulate the problem as one of offline reinforcement learning (Huo et al., 2023).

## 3. Spectral Analysis of Robot Actions

### 3.1. Problem Statement and Notation

We study imitation learning for robotic manipulation, modeled as a Markov Decision Process (MDP). At each time step $t$, the robot receives an observation $o_t$ (e.g., multi-view RGB images) together with a language instruction $l$, and executes a continuous action $a_t \in \mathbb{R}^d$. A policy $\pi(a \mid o, l)$ is trained to predict actions conditioned on both observation and language.

Given a dataset of demonstration trajectories

$$\mathcal{D} = \{\tau_i\}_{i=1}^N, \qquad \tau_i = \{(o_0, a_0), \ldots, (o_T, a_T)\},$$

where each trajectory corresponds to a successful execution of the instructed task. The objective of imitation learning is to learn a policy whose action distribution matches that of the demonstrations,

$$P_{\text{pred}}(a \mid o, l) \approx P_{\text{gt}}(a \mid o, l).$$

**Temporal spectral representation.** To describe temporal structure, we consider a generic discrete time series $x = (x_0, x_1, \ldots, x_{K-1}), x_k \in \mathbb{R}$. We represent this sequence in the temporal frequency domain using the discrete cosine transform (DCT). Let $\boldsymbol{F} \in \mathbb{R}^{K \times K}$ denote the orthonormal DCT basis. The forward transform produces spectral coefficients

$$\omega = \boldsymbol{F}x,$$

with individual coefficients given by

$$\omega_k = \sum_{n=0}^{K-1} x_n \cos\left(\frac{\pi}{K}\left(n + \tfrac{1}{2}\right)k\right), \qquad k = 0, \dots, K-1.$$

The inverse transform reconstructs the original sequence as $x = \boldsymbol{F}^{-1}\omega$, which is exact since the DCT basis is orthonormal.

In practice, this transform is applied independently to each dimension of an action sequence, providing an invertible reparameterization that makes temporal structure at different scales explicit.

### 3.2. Frequency-Domain Intervention via Coefficient Truncation

Building on the temporal spectral representation introduced above, we investigate the role of different frequency components in robot action trajectories. We perform controlled frequency-domain interventions on both simulated and real-robot demonstrations by modifying spectral coefficients and replaying the reconstructed actions.

Concretely, given a demonstrated action trajectory, we divide the sequence into fixed-length chunks of length $K$ and apply the discrete cosine transform (DCT) independently to each action dimension, yielding a spectral representation $c = \boldsymbol{F}a$, where $a \in \mathbb{R}^{K \times d}$ denotes the original action chunk and $c \in \mathbb{R}^{K \times d}$ its corresponding frequency coefficients.

To probe the functional role of different frequency bands, we perform controlled truncation of high-frequency components. For a truncation level $0 < \lambda < K$, we zero out the highest $\lambda$ frequency coefficients:

$$\tilde{c}_i = \begin{cases} c_i, & i \leq K - \lambda, \\ 0, & i > K - \lambda. \end{cases}$$

The modified action chunk is reconstructed via inverse DCT, $\tilde{a} = F^{-1}\tilde{c}$, and replayed in the environment without further modification. By varying the number of retained low-frequency components while keeping observations fixed, this procedure isolates the causal effect of temporal frequency content on task execution.

### 3.3. Real-Robot Dart Insertion Task

We apply this analysis to a real-robot dart insertion task, where the robot aims to hit the center of a circular target (Fig. 2). Successful execution requires both a correct coarse trajectory toward the target and precise alignment during the final insertion phase, making the task suitable for studying the role of temporal frequency components.

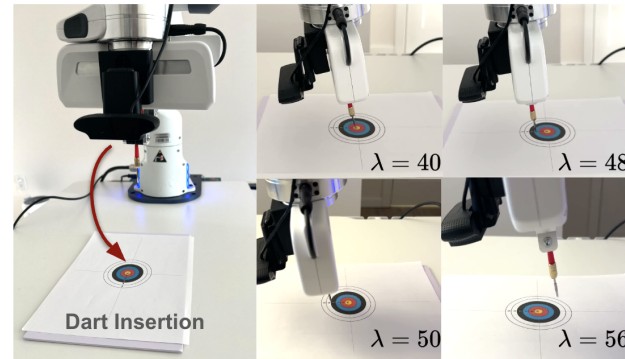

*Figure 2.* Action reconstruction under different frequency cutoffs $\lambda$ in the real-robot dart insertion task. As $\lambda$ decreases, coarse motion toward the target is preserved while fine alignment and contact accuracy degrade, highlighting the role of high-frequency components in precision execution.

We collect a single successful demonstration and analyze the action sequence using a fixed chunk size of $K = 64$. By progressively zeroing out high-frequency coefficients beyond a cutoff $\lambda$, we replay reconstructed trajectories with different frequency content (Fig. 2). When a large portion of high-frequency components is preserved (e.g., $\lambda = 40$), the reconstructed motion closely matches the original behavior and accurately hits the target center. As $\lambda$ decreases, the robot continues to reach the target region but exhibits increasing misalignment, resulting in off-center insertions. When only low-frequency components remain, the robot follows the correct coarse approach but fails to achieve contact. These results show that low-frequency components govern global reaching motion, while high-frequency components are critical for precise alignment and successful insertion.

## 4. Method

### 4.1. Decomposition Across Temporal Scales

The frequency-domain analysis in the previous section reveals a clear separation in how robot actions contribute to task execution: low-frequency components govern global reaching behavior, while high-frequency components are responsible for precise alignment and contact. This observation motivates a structural hypothesis about manipulation.

Specifically, observation and language primarily determine the coarse trajectory of a task. Once this trajectory is selected, the robot largely knows where to move and which region of the scene to interact with. Remaining refinements—such as small pose adjustments or contact timing—are generated relative to the realized motion and depend only weakly on the language instruction. This structure is common in manipulation, where committing to a motion context constrains subsequent behavior to a limited set of interaction-specific actions.

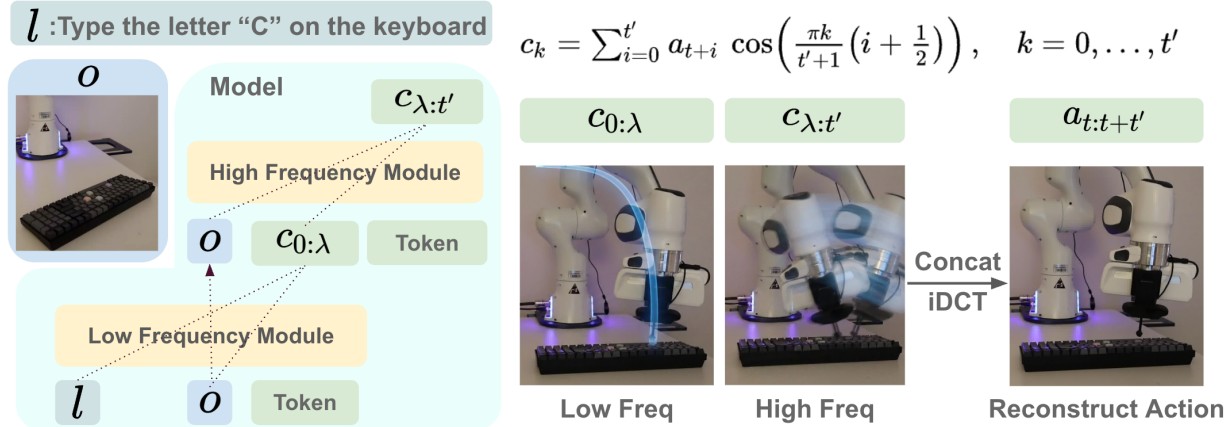

$$c_k = \sum_{i=0}^{t'} a_{t+i} \cos\left(\frac{\pi k}{t'+1}\left(i + \frac{1}{2}\right)\right), \quad k = 0, \dots, t'$$

*Figure 3.* Overview of the proposed hierarchical spectral policy. Given observation $o$ and language instruction $l$, the policy first predicts low-frequency action components that capture coarse task-level motion. Conditioned on the realized low-frequency trajectory, a second module predicts high-frequency corrective components. The final action sequence is reconstructed by concatenating frequency coefficients and applying the inverse discrete cosine transform (iDCT). This causal coarse-to-fine structure enables robust learning of fine-grained execution under noisy demonstrations.

Motivated by this observation, we model action generation as a coarse-to-fine process across temporal scales. Given an action chunk $a_{t:t+t'}$, we factorize its distribution as

$$p(a_{t:t+t'} \mid o_t, l) = p(a^{\text{coarse}} \mid o_t, l)\, p(a^{\text{fine}} \mid o_t, a^{\text{coarse}}). \tag{1}$$

In contrast, standard chunk-based and autoregressive policies predict fine-scale actions without explicitly modeling coarse motion, which often leads to ambiguous supervision and averaged refinement behavior (Fig. 4). This highlights the need for an action representation that enables a clear separation between coarse structure and fine-grained corrections.

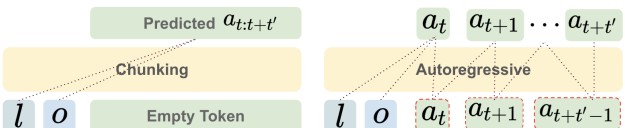

*Figure 4.* Baseline action prediction paradigms. **Left:** chunk-based joint prediction. **Right:** autoregressive prediction. Both lack an explicit representation of coarse temporal structure, motivating our coarse-to-fine formulation.

### 4.2. Action Representation in the Spectral Domain

For the coarse-to-fine factorization in Eq. (1) to be effective, the action representation must make temporal structure at different resolutions explicit. In the time domain, however, all timesteps are treated uniformly, which entangles global motion with local refinements.

We therefore represent actions in the spectral domain using the discrete cosine transform (DCT), following (Pertsch et al., 2025). The DCT provides an invertible and orthogonal

transformation that reorganizes a temporal signal according to frequency.

Given an action chunk $a_{t:t+t'} \in \mathbb{R}^{K \times d}$, the transform is applied independently to each action dimension. Let $F \in \mathbb{R}^{K \times K}$ denote an orthonormal DCT basis: $c_t = F a_{t:t+t'}$, and $a_{t:t+t'} = F^{-1} c_t$.

Because $F$ is orthonormal, the transformation preserves Euclidean distance, $\|\hat{a} - a\|_2^2 = \|\hat{c} - c\|_2^2$, and therefore constitutes an exact reparameterization of the original action sequence.

Importantly, the spectral domain naturally aligns with the coarse-to-fine structure assumed in Eq. (1). Under a fixed control frequency, low-frequency coefficients correspond to slowly varying motion over long temporal horizons, while high-frequency coefficients capture rapid, short-timescale variations. Equivalently, a coefficient at frequency $k$ reflects action variation at a temporal scale approximately proportional to $1/k$. As a result, the spectral representation orders action structure from coarse to fine, providing a principled basis for separating global motion planning from local execution refinement.

### 4.3. Causal Spectral Policy

Formally, we propose **CSP**, a causal temporal decomposition policy in the spectral domain. Given an action chunk $a_{t:t+t'}$ and its frequency coefficients $c = F a_{t:t+t'}$, modeling actions in the time domain is equivalent to modeling their spectral representation: $p(a_{t:t+t'} \mid o_t, l) \equiv p(c \mid o_t, l)$, since the transform $F$ is invertible.

We partition the coefficients into coarse and fine components.

For some cutoff parameter $\lambda \in (0, t'), \lambda \in \mathbb{N}$,

$$c = \begin{bmatrix} c^{\text{low}} \\ c^{\text{high}} \end{bmatrix} = \begin{bmatrix} c_{0:\lambda} \\ c_{\lambda:t'} \end{bmatrix},$$

and model them using two separate predictors with a causal dependency:

$$p(c \mid o_t, l) = p_{\text{low}}(c^{\text{low}} \mid o_t, l) \, p_{\text{high}}(c^{\text{high}} \mid o_t, c^{\text{low}}). \quad (2)$$

This factorization reflects the structure of manipulation. Observation and language primarily determine the coarse trajectory, while fine-scale execution is generated relative to that trajectory. The resulting coarse-to-fine architecture, illustrated in Fig. 3, allows global task intent to be decided first, followed by local execution refinement.

**Training.** Following the factorization in Eq. (2), we train two predictors for coarse and fine spectral components. Given ground-truth coefficients $c = [c^{\text{low}}, c^{\text{high}}]$, the training objective is

$$\mathcal{L} = \|\hat{c}^{\text{low}} - c^{\text{low}}\|_2^2 + \|\hat{c}^{\text{high}} - c^{\text{high}}\|_2^2, \quad (3)$$

where $\hat{c}^{\text{low}} = p_{\text{low}}(o_t, l)$ and $\hat{c}^{\text{high}} = p_{\text{high}}(o_t, \text{sg}(\hat{c}^{\text{low}}))$. The stop-gradient operator $\text{sg}(\cdot)$ prevents fine-scale supervision from influencing coarse motion learning, enforcing the intended causal dependency.

# 5. Experiments

Our experiments are designed to test the following hypotheses:

1. Robot manipulation actions exhibit structured organization across temporal frequencies, with different components playing distinct functional roles in task execution.
2. Counterfactual interventions on the conditioning structure can reveal whether the proposed coarse-to-fine factorization captures a directional dependency between low-frequency task intent and high-frequency execution refinements.
3. Explicitly modeling this structure improves imitation learning performance compared to standard chunk-based and autoregressive policies, particularly on long-horizon and precision-sensitive tasks.
4. Temporal abstraction alone is insufficient; incorporating frequency-domain structure is necessary to capture fine-grained execution behavior.
5. Modeling action structure across temporal frequencies improves robustness to execution noise in both teleoperated demonstrations and real-robot data.

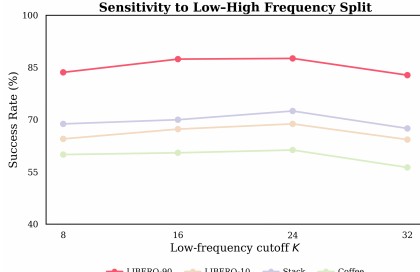

*Figure 5.* **Sensitivity to the low–high frequency split.** Performance remains stable across a wide range of low-frequency cutoffs, indicating low sensitivity to the exact partition.

## 5.1. Ideal Simulation Experiments

### 5.1.1. EXPERIMENTAL SETUP

We evaluate our method in simulation to assess whether modeling spectral structure improves imitation learning under clean supervision. Experiments are conducted on two standard benchmarks: **LIBERO** (Liu et al., 2023) (LIBERO-90 and LIBERO-10), which emphasizes long-horizon, language-conditioned tasks, and **MimicGen** (Mandlekar et al., 2023), which focuses on precision-sensitive behaviors such as stacking and threading. To analyze sensitivity to temporal horizon, we vary the action chunk size while keeping the execution horizon fixed.

For each task and chunk length $K$, we determine the coarse–fine frequency split using a data-driven heuristic. We compute the DCT of demonstrated action chunks, average squared coefficients across demonstrations and action dimensions to obtain an empirical power spectrum, and select the smallest index whose cumulative energy exceeds 90%. Across both benchmarks, this typically assigns the lowest ~30% of frequency coefficients to the coarse component. Performance remains stable over a wide range of cutoffs (approximately 20%–40%), as shown in Fig. 5, indicating that **CSP** is not sensitive to the exact frequency partition.

### 5.1.2. BASELINES & ABLATIONS

We compare against baselines that isolate temporal abstraction, frequency representation, and hierarchical structure, as summarized in Figure 7. **Action Binning** tests time-domain hierarchy. **Frequency Autoregressive** tests spectral sequential prediction without explicit coarse–fine conditioning. We also include standard imitation learning baselines (**ACT** (Fu et al., 2024), **BAKU** (Haldar et al., 2024), and **Diffusion Policy** (Chi et al., 2023)) that predict actions directly in the time domain. For ablation, **Frequency Diffusion** and **No Hierarchy** remove hierarchical conditioning while retaining spectral modeling. Together, these variants disentangle temporal abstraction, spectral representation, and causal coarse–fine decomposition.

*Table 1.* **Simulation Results under Different Action Chunk Sizes.** Success rate (%) across LIBERO and MimicGen-style tasks, averaged across 2 seeds. While longer chunk lengths are preferred for multimodal consistency, concurrent methods only work with short chunk lengths in precision tasks. This result shows that CSP is better at preserving performance while increasing look-ahead during prediction.

| Method | Libero-90 | Libero-10 | Stack | Stack3 | Coffee | Square | Threading | MimicGen Mean |
|---|---|---|---|---|---|---|---|---|
| **Chunk Size = 16** | | | | | | | | |
| ACT | 90.7 | 70.5 | 81.3 | 45.0 | 72.5 | 37.5 | 21.3 | 51.5 |
| BAKU | 86.1 | 57.4 | 86.3 | 35.0 | 70.0 | 31.3 | 26.3 | 49.8 |
| DP-CNN | 83.0 | 64.6 | 88.8 | 58.8 | 58.8 | **52.5** | 33.8 | 58.5 |
| DP-Transformer | 77.0 | 60.6 | 82.5 | 52.5 | 58.8 | 38.8 | 36.3 | 53.8 |
| Action Binning | **92.1** | 73.8 | 81.3 | **60.0** | 82.5 | 47.5 | 17.5 | 57.8 |
| Freq-Autoregressive | 90.9 | 76.8 | 88.8 | 45.0 | 67.5 | 43.8 | 42.5 | 57.5 |
| **CSP(Ours)** | 91.9 | **80.8** | **92.5** | 56.3 | **83.8** | 48.8 | **46.3** | **65.5** |
| **Chunk Size = 64** | | | | | | | | |
| ACT | 85.7 | 48.6 | 0.0 | 15.0 | 21.3 | 3.8 | 5.0 | 9.0 |
| BAKU | 75.4 | 35.1 | 48.8 | 5.0 | 8.8 | 15.0 | 8.8 | 17.3 |
| DP-CNN | 82.7 | 46.4 | 73.8 | 31.3 | 35.0 | 36.3 | 32.5 | 41.8 |
| DP-Transformer | 79.1 | 50.0 | 65.0 | 21.3 | 26.3 | 21.3 | 22.5 | 31.3 |
| Action Binning | 84.9 | 59.8 | 62.5 | 17.5 | 47.5 | 23.8 | 7.5 | 31.8 |
| Freq-Autoregressive | 85.9 | 60.5 | **77.5** | 10.0 | 48.8 | 16.3 | 7.5 | 32.0 |
| **CSP(Ours)** | **87.6** | **68.8** | 72.5 | **33.8** | **61.3** | **43.8** | **38.8** | **50.0** |

*Table 2.* **Ablation Study.** Evaluating the effects of hierarchy and frequency modeling under simulated teleoperation noise.

| Method | Stack | Coffee | Libero-10 |
|---|---|---|---|
| Frequency Diffusion | **81.3** | 56.3 | 61.3 |
| No Hierarchy | 73.8 | 77.5 | 56.3 |
| **CSP** | 75.0 | **82.5** | **74.0** |

| | Libero-90 (Set A) | | |
|---|---|---|---|
| | **No** | **Small** | **Large** |
| Frequency Diffusion | 92.5 | 78.5 | 73.0 |
| No Hierarchy | 95.5 | 85.5 | 81.5 |
| **CSP** | **97.0** | **92.0** | **87.5** |

| | Libero-90 (Set B) | | |
|---|---|---|---|
| | **No** | **Small** | **Large** |
| Frequency Diffusion | 94.0 | 77.5 | 63.8 |
| No Hierarchy | **96.0** | 81.5 | 72.5 |
| **CSP** | 94.0 | **84.0** | **80.0** |

### 5.1.3. RESULTS AND ANALYSIS

Table 1 shows that explicitly modeling a causal spectral coarse-to-fine structure significantly improves imitation learning across both long-horizon and precision-sensitive tasks. On LIBERO, **CSP** consistently matches or outperforms all baselines at both chunk sizes, achieving 91.9% / 80.8% success on LIBERO-90 / LIBERO-10 at chunk size 16 and maintaining strong performance at chunk size 64 (87.6% / 68.8%), while standard chunk-based and autoregressive policies degrade substantially as the temporal horizon increases.

The advantage is more pronounced on MimicGen, where task success depends on accurate alignment and timing. At chunk size 16, **CSP** achieves the highest mean success rate (65.5%), and at chunk size 64 it degrades gracefully

to 50.0%, whereas ACT and BAKU collapse below 20%. Action Binning provides moderate gains at short horizons but deteriorates sharply at longer horizons (31.8% mean), indicating that temporal abstraction alone is insufficient. Frequency-Autoregressive methods benefit from spectral representations but remain consistently below **CSP** without explicit coarse–fine conditioning. These results support Hypotheses 3 and 4.

Table 2 further clarifies the source of these gains. Removing the hierarchical structure or the causal factorization both leads to noticeable performance drops, particularly under noisy supervision. Only the full **CSP** model maintains strong performance across settings, supporting Hypotheses 1 and 5 and confirming that structured spectral decomposition is critical for reliable manipulation.

### 5.1.4. COUNTERFACTUAL INTERVENTIONS

To further examine Hypothesis 2, we perform counterfactual interventions on the conditioning structure of **CSP**. While the previous results show that spectral hierarchy improves policy learning, this analysis tests whether the proposed low-to-high frequency ordering reflects a directional dependency rather than an arbitrary hierarchical design. We construct variants that preserve the same spectral action representation but alter the dependency between coarse and fine components: removing language from the low-frequency predictor, removing low-frequency conditioning from the high-frequency predictor, and reversing the prediction order by generating high-frequency components before low-frequency components.

Table 3 shows a clear asymmetric dependency. Removing language from the low-frequency module causes a large

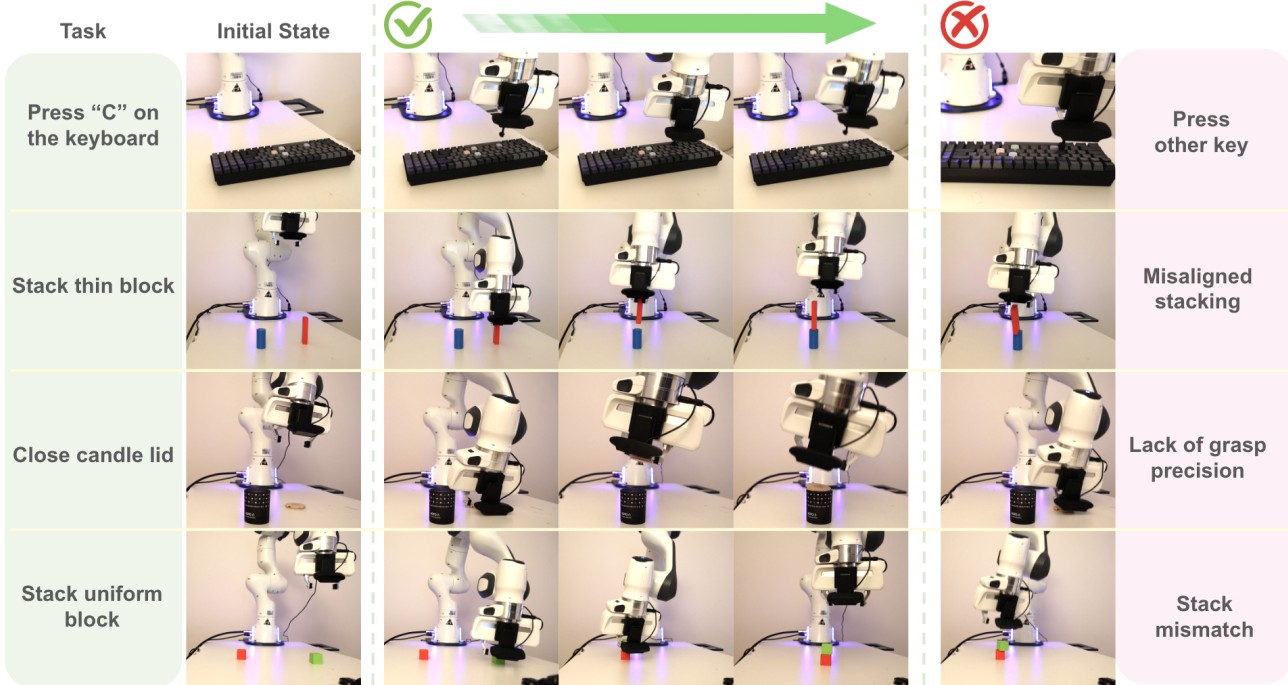

*Figure 6.* Real-robot qualitative results across four manipulation tasks. For each task, we show the initial state, successful executions by **CSP** (middle), and representative failure cases from baseline policies (right). Tasks require precise alignment and contact, including keyboard pressing, block stacking, and lid closing, highlighting the importance of accurate fine-grained execution.

| Method | Low-frequency condition | High-frequency condition | L90 K=16 | L90 K=32 | L10 K=16 | L10 K=32 |
|---|---|---|---|---|---|---|
| No language in coarse | $o$ | $o, l, c^{low}$ | 22.2 | 18.2 | 64.5 | 61.0 |
| No coarse in fine | $o, l$ | $o$ | 86.2 | 84.2 | 65.0 | 51.5 |
| **CSP** (Ours) | $o, l$ | $o, l, c^{low}$ | **91.6** | **90.4** | **81.0** | **75.0** |
| Reverse order | $c^{high}, o$ | $o, l$ | 49.6 | 46.0 | 35.5 | 55.5 |
| Reverse order + language | $c^{high}, o, l$ | $o, l$ | 80.3 | 66.8 | 49.5 | 46.0 |

*Table 3.* Counterfactual intervention analysis. **CSP** (Ours) uses the proposed low-to-high conditioning structure and achieves the best performance. Removing coarse task conditioning or reversing the prediction order degrades success rates, supporting the proposed asymmetric coarse-to-fine structure.

drop, indicating that task-level intent is primarily expressed through coarse motion. Removing low-frequency conditioning from the high-frequency module also degrades performance, showing that fine corrections depend on the realized coarse trajectory. Most importantly, reversing the dependency direction performs substantially worse than the proposed ordering, suggesting that the low-to-high frequency factorization is not interchangeable with a high-to-low alternative. These results provide counterfactual evidence that the causal ordering in **CSP** is an empirically useful structural bias, rather than merely an architectural choice.

## 5.2. Real Robot Evaluation

Real robot experiments were conducted on a Franka Emika Panda equipped with a fixed and a wrist-mounted Logitech RGB camera. Our real evaluation tasks can be decomposed into three categories, each with their own failure modalities:

| Task | Baselines | | | Ours |
|---|---|---|---|---|
| | DP | Baku | Action Binning | CSP |
| Press Enter Key | 1/10 | 8/10 | **10/10** | 10/10 |
| Press C | 0/10 | 3/10 | 5/10 | **8/10** |
| Stack uniform block | 6/10 | 6/10 | **9/10** | 9/10 |
| Stack thin block | 2/10 | 3/10 | 6/10 | **9/10** |
| Close candle lid | 3/10 | 5/10 | 3/10 | **7/10** |

*Table 4.* Real-robot success rates (successes / 10 trials). Tasks emphasize precise alignment and contact. Best result per task is highlighted in bold.

typing, stacking, and component assembly. Typing tasks are defined by pressing the commanded key while avoiding neighboring keys, requiring precise alignment to avoid the rigid failure states. For stacking tasks, blocks must be axially aligned. Fig. 6 includes example failure modes for each task category.

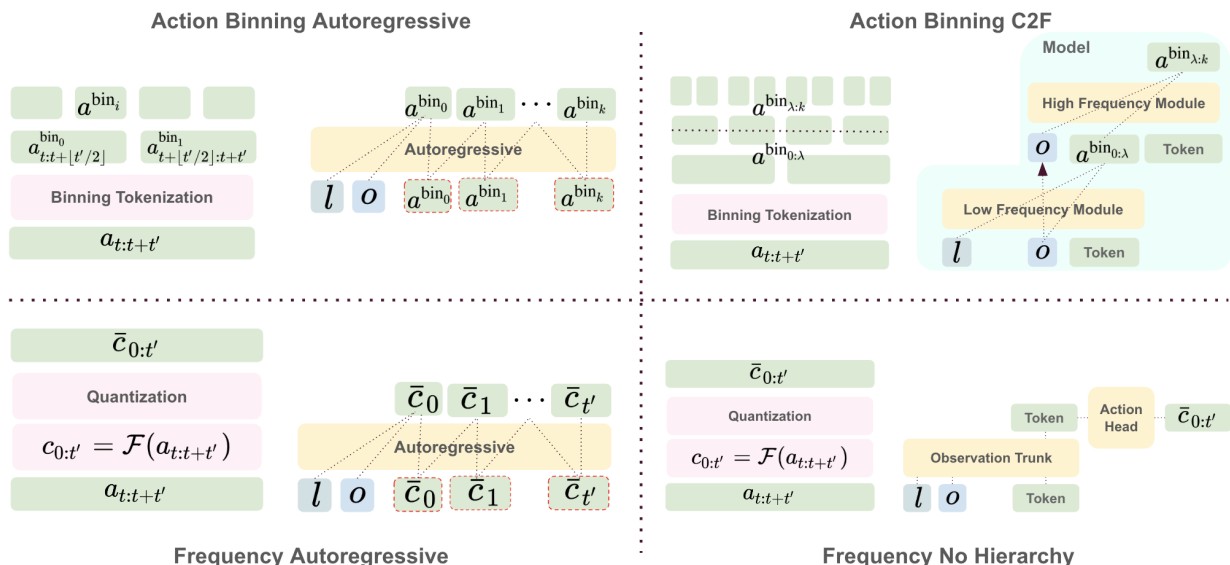

*Figure 7.* Baseline and ablation architectures used to isolate temporal abstraction, spectral representation, and coarse-to-fine conditioning.

Table 4 and Fig. 6 summarize our real-robot results. Across all tasks, **CSP** consistently improves execution accuracy, achieving the highest success on all precision-critical tasks. On keyboard tapping, our method attains 80% success on *Press C* and 100% on *Press Enter*, outperforming all baselines; the *Enter* key is physically larger and therefore easier to align with, leading to higher success across methods. On stacking, performance diverges sharply by task difficulty: for *uniform block stacking*, all methods—including naïve behavior cloning—achieve high success due to clean demonstrations and an average teleoperation success rate of approximately 90%. In contrast, *thin block stacking* is substantially more challenging, with only ∼50% teleoperation success; demonstrations are longer and contain frequent high-frequency corrective motions. These interventions cause baseline methods to fail due to misalignment while our method achieves a 90% success rate. For *closing a candle lid*, baseline methods suffer from unstable grasping and low-precision alignment, whereas our method produces more reliable contact and closure, achieving the highest success rate. Overall, these results show that coarse-to-fine spectral decomposition is especially effective for precision-sensitive and noisy manipulation, with gains in simulation translating consistently to real-world performance.

## 5.3. Simulated Human Noise Injection

To study robustness under realistic supervision, we inject human-like execution noise into simulated demonstrations. This allows us to systematically evaluate how different policy structures degrade under noisy training data, and whether **CSP** better preserves task-critical behavior.

### 5.3.1. NOISE EXPERIMENT SETUP

To evaluate robustness to noisy demonstrations, we conduct controlled noise injection experiments in simulation. While real-robot data naturally contains execution variability, simulation allows precise control over noise structure and magnitude. In practice, we evaluate two multi-task LIBERO-90 subsets (Set A and Set B) comprised of 10 tasks each and three single-task datasets, the mean of which is presented as Set C (individual tasks in Appendix). For each setting, models are trained under three conditions: *no noise*, *moderate noise*, and *high noise*, and evaluated using clean rollouts.

**Human-inspired noise model.** Instead of i.i.d. perturbations, we inject structured noise that approximates human teleoperation behavior. Given a clean action $a_t$, the executed action is

$$\tilde{a}_t = a_t + \lambda(t)(1-m_t)\left[(s_0+k|a_t|)\odot n_t + \sum_j \mathbf{1}[t \in W_j]\eta_t^{(j)}\right].$$
(4)

where $n_t$ follows an AR(1) process, $n_t = \rho n_{t-1} + \epsilon_t$, and $\epsilon_t \sim \mathcal{N}(0, \sigma^2)$, modeling smooth execution drift. The scaling $(s_0 + k|a_t|)$ captures signal-dependent motor variability, while burst terms $\eta_t^{(j)}$ injected over short windows $W_j$ represent intermittent corrective motions.

The phase-dependent factor $\lambda(t)$ reduces noise near task completion, and $m_t$ optionally disables perturbations at critical events to ensure task success.

**Noise regimes.** We evaluate two levels of simulated teleoperation noise. **Small noise** introduces mild temporally

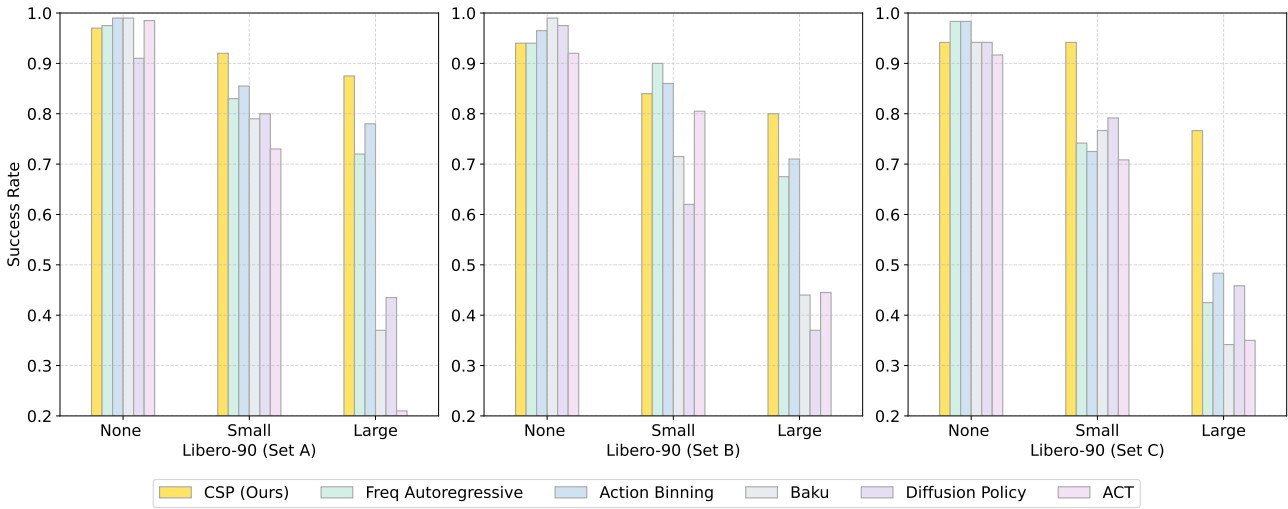

*Figure 8.* **Libero Noise Injection Results** For single tasks and subsets of libero_90, **CSP** is more resistant to noise injection than all evaluated baselines. The noise model herein models real-world teleoperation noise, and this result showcases that while previous architecture capture cleaner data better, but perform rather poorly in prescence of high-frequency noise present in real data.

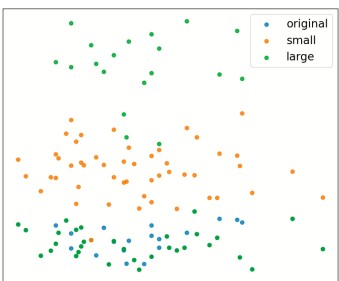

*Figure 9.* PCA visualization of action trajectories under different noise levels. Small noise causes limited dispersion, while large noise leads to substantially greater variation

correlated drift ($\rho \approx$ 0.4–0.6) with limited stochastic variation and weak corrective bursts. **Large noise** increases temporal correlation ($\rho \approx$ 0.6–0.9) and burst magnitude, resulting in substantially noisier execution. In both regimes, noise intensity decays toward task completion via a phase-dependent schedule $\lambda(t)$. As illustrated in Fig. 8, injected noise primarily affects short-timescale action components while largely preserving low-frequency task intent. Full parameter settings are provided in Appendix 7.1.4.

### 5.3.2. NOISE EXPERIMENTAL RESULTS & ANALYSIS

We apply the proposed human noise injection framework to Libero demonstrations to evaluate robustness under realistic execution variability. Low- and high-noise regimes introduce increasing short-timescale perturbations that primarily affect fine-grained action components. As shown in Fig. 8, **CSP** consistently achieves higher success rates than all baselines across three individual tasks and two multi-task subsets, and degrades substantially less under high-noise conditions. Ablation results in Table 2 further indicate that both hierarchical structure and causal frequency modeling are critical for robustness, as removing either component leads to significantly larger performance drops under noise.

These results directly support our hypotheses. The strong

performance gap under noise validates Hypothesis 5, showing that explicitly modeling action structure across temporal frequencies improves robustness. Moreover, the failure of time-domain baselines under increasing noise aligns with Hypothesis 4, indicating that temporal abstraction alone is insufficient without frequency-aware modeling. Together with the clean-setting results, these findings reinforce Hypotheses 1 and 3, demonstrating that spectral structure is both behaviorally meaningful and practically beneficial for stable manipulation learning.

## 6. Conclusion

We identify a structured decomposition in robot manipulation actions, where task-level motion and execution-level refinements play distinct roles in successful behavior, and show that this structure naturally emerges in the spectral domain through low- and high-frequency components. Motivated by this insight, we propose **CSP**, a causal spectral policy that generates actions via an explicit coarse-to-fine process. Extensive simulation and real-robot experiments demonstrate that modeling this structure improves performance on precision-sensitive tasks and substantially enhances robustness to realistic teleoperation noise.

## Impact Statement

Robotic manipulation is a fundamental capability for autonomous systems in domains such as manufacturing, service robotics, and assistive technologies. However, existing imitation learning methods often struggle with long-horizon control, precision-sensitive behaviors, and robustness to noisy demonstrations, limiting real-world deployment.

This work contributes a structured perspective on robot action generation by identifying semantic organization across temporal frequencies and proposing a causal spectral coarse-to-fine policy. By improving learning stability and robustness under realistic teleoperation noise, our approach enables more reliable learning from human demonstrations and reduces dependence on idealized data collection.

The potential positive impacts include improved scalability of robot learning systems, greater tolerance to imperfect supervision, and broader applicability of imitation learning in real-world environments. We do not identify risks specific to this method beyond those common to autonomous robotic systems. As with all learning-based control approaches, responsible deployment requires appropriate safety validation, human oversight, and domain-specific risk assessment.

## Acknowledgment

We are grateful to Ruoshi Liu for his insightful advice and fruitful discussions, and to the PairLab members at Georgia Tech for their helpful feedback and peer review. This work was supported in part by the Stephen Fleming Early Career Grant, seed grants from IMS, Mechanical Engineering WIN, and IRIM at Georgia Tech, as well as GTRI.

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

# 7. Appendix

## 7.1. Human-Inspired Noise Injection

### 7.1.1. MOTIVATION AND BACKGROUND

**Motivation.** When humans teleoperate a robot, executed actions deviate systematically from an ideal, noise-free policy. Neuroscience and motor-control studies show that this variability is not i.i.d. noise, but exhibits structured properties governed by human motor planning and feedback control. In particular, human execution noise is characterized by three key phenomena:

- **Signal-dependent motor noise:** the variance of motor commands increases with command magnitude, making larger and faster movements intrinsically noisier than small corrective actions (Harris & Wolpert, 1998b).

- **Temporally correlated execution drift:** motor variability is smooth and correlated over time due to muscle dynamics and neural control, resulting in low-frequency drift rather than frame-independent jitter (Van Beers et al., 2004; Faisal et al., 2008).

- **Intermittent corrective submovements:** under optimal feedback control, humans tolerate variability during motion but generate brief, high-frequency corrections when precision becomes critical, particularly near task completion (Todorov & Jordan, 2002b).

### 7.1.2. NOISE MODEL FORMULATION

To model these effects, we define the executed (noisy) action $\tilde{a}_t \in \mathbb{R}^d$ as a structured perturbation of an ideal simulator action $a_t$:

$$\tilde{a}_t = a_t + \lambda(t)\,(1 - m_t)\Big[(s_0 + k|a_t|) \odot n_t + \sum_j \mathbf{1}[t \in W_j]\,\eta_t^{(j)}\Big], \tag{5}$$

where all operations are elementwise unless otherwise stated.

**Signal-dependent execution variability.** The term $(s_0 + k|a_t|) \in \mathbb{R}_+^d$ implements signal-dependent motor noise. Here, $s_0$ denotes a per-dimension baseline noise level, while $k$ scales noise proportionally to the action magnitude. This formulation reflects the classic observation that larger motor commands induce greater execution variability (Harris & Wolpert, 1998a).

**Temporally correlated execution drift.** The process $n_t \in \mathbb{R}^d$ models smooth execution drift and evolves according to an AR(1) process:

$$n_t = \rho n_{t-1} + \epsilon_t, \qquad \epsilon_t \sim \mathcal{N}(0, \Sigma_\epsilon), \tag{6}$$

where $\rho \in [0, 1)$ controls temporal correlation. This produces low-frequency deviations consistent with empirical

findings that human execution noise is temporally correlated rather than white (Van Beers et al., 2004; Faisal et al., 2008). Combined with the signal-dependent scaling, this yields structured, amplitude-aware execution noise.

**Intermittent corrective bursts.** Human demonstrations often contain brief, high-frequency corrective actions during precision-critical phases (e.g., alignment before grasp or insertion). We model these as localized burst perturbations $\eta_t^{(j)} \sim \mathcal{N}(0, \Sigma_{\text{burst}})$ injected within short temporal windows $W_j = [s_j, s_j + L_j]$:

$$\sum_j \mathbf{1}[t \in W_j]\,\eta_t^{(j)}. \tag{7}$$

These bursts correspond to feedback-driven corrections under optimal feedback control (Todorov & Jordan, 2002b), introducing high-frequency variability without corrupting global task structure.

**Task-phase modulation and safety.** The scalar $\lambda(t) \in (0, 1]$ modulates noise magnitude as a function of task progress, typically decaying toward zero near task completion. The binary mask $m_t \in \{0, 1\}$ disables noise at critical timesteps (e.g., around gripper closure). Together, these mechanisms ensure that injected noise preserves task intent and success while concentrating variability in mid-trajectory and precision phases.

**Summary** Overall, human execution noise is modeled as

*signal-dependent, temporally correlated drift + sparse feedback-driven corrective bursts.*

This structure primarily perturbs high-frequency components of the action sequence while leaving low-frequency, task-level motion largely intact.

### 7.1.3. SIMULATED HUMAN TELEOPERATION NOISE

In addition to real-robot experiments, we apply the same noise model to simulated datasets to study performance degradation under corrupted demonstrations. Unlike naïve i.i.d. perturbations, our model explicitly captures both smooth motor drift and intermittent corrective actions.

Given a clean action $a_t$, the executed action $\tilde{a}_t$ follows Eq. (5). The AR(1) drift process introduces smooth deviations commonly observed during continuous teleoperation, while burst perturbations model brief corrective submovements. Task-phase modulation via $\lambda(t)$ and masking via $m_t$ ensure that all noisy trajectories remain task-completing.

Because both drift and corrective bursts concentrate variability at short temporal scales, injected noise predominantly affects high-frequency components of the trajectory. Low-

frequency coefficients encoding global task motion remain comparatively stable.

### 7.1.4. NOISE REGIMES

To simulate varying levels of teleoperation difficulty, we define two noise regimes:

**Low-noise regime.** This setting models mild human variability. Drift parameters are sampled from $\rho \in [0.40, 0.65]$ and $\sigma \in [0.22, 0.35]$. Signal-dependent scaling uses $s_0 \in [0.05, 0.10]$ and $k \in [0.15, 0.28]$. Corrective bursts occur sparsely, with 0–4 windows of 2–4 timesteps and burst magnitude $\eta \sim \mathcal{N}(0, \sigma_b^2)$ where $\sigma_b \in [0.06, 0.12]$. Noise intensity decays with $\lambda(t) \in [0.25, 0.40]$, and per-dimension perturbations are clipped to $[0.22, 0.36]$.

**High-noise regime.** This regime simulates challenging teleoperation with substantial execution instability. Drift correlation and magnitude increase to $\rho \in [0.60, 0.90]$ and $\sigma \in [0.45, 0.75]$. Signal-dependent variability is amplified using $s_0 \in [0.12, 0.22]$ and $k \in [0.32, 0.55]$. Burst corrections are more frequent and stronger, with 0–6 windows of 2–5 timesteps and $\sigma_b \in [0.16, 0.32]$. Noise decays later in the trajectory via $\lambda(t) \in [0.15, 0.30]$, and clipping bounds increase to $[0.45, 0.75]$ per dimension.

**Spectral characteristics.** Across both regimes, injected noise predominantly perturbs high-frequency components of the action trajectory, while low-frequency coefficients encoding global task-level motion remain stable.

### 7.1.5. MOTIVATION: COARSE-TO-FINE LEARNING UNDER NOISY DEMONSTRATIONS

The coarse-to-fine factorization of CSP becomes particularly important under realistic demonstration noise. In teleoperated data, execution variability is unevenly distributed across temporal scales: global task-level motion is often consistent across demonstrations, while fine-scale corrective behavior exhibits substantial variability.

Let $a_{t:t+t'}$ denote an action chunk and $c = Fa_{t:t+t'}$ its spectral representation. Under human teleoperation, the executed trajectory is corrupted by noise,

$$\tilde{a} = a + \varepsilon, \qquad \tilde{c} = c + \eta, \quad \eta = F\varepsilon.$$

Due to temporal correlation in execution drift and delayed human feedback, this noise primarily affects short-timescale variations, yielding

$$\eta^{\text{low}} \approx 0, \qquad \eta^{\text{high}} \text{ large},$$

where $\eta^{\text{low}}$ and $\eta^{\text{high}}$ denote the low- and high-frequency components of the noise, respectively.

Under standard imitation learning, training minimizes squared error on noisy demonstrations,

$$\mathbb{E}\big[\|\hat{c} - \tilde{c}\|_2^2\big] = \mathbb{E}\big[\|\hat{c}^{\text{low}} - c^{\text{low}}\|_2^2\big] + \mathbb{E}\big[\|\hat{c}^{\text{high}} - (c^{\text{high}} + \eta^{\text{high}})\|_2^2\big].$$

When the variance of $\eta^{\text{high}}$ is large, optimization becomes dominated by noisy high-frequency supervision. As a result, the learned predictor collapses toward the conditional mean,

$$\hat{c}^{\text{high}}(o_t, l) = \mathbb{E}[c^{\text{high}} \mid o_t, l],$$

which suppresses task-critical corrective behavior and yields overly smooth executions.

In contrast, CSP conditions fine-scale prediction on the realized coarse trajectory,

$$p(c^{\text{high}} \mid o_t, l) \ \rightarrow \ p(c^{\text{high}} \mid o_t, c^{\text{low}}).$$

By anchoring high-frequency refinement to a specific low-frequency motion context, this factorization substantially reduces ambiguity in high-frequency supervision. Consequently, predictable corrective structure can be learned even when demonstrations contain significant execution noise—an essential property for manipulation tasks requiring precise alignment and timing.

### 7.2. Zero-Out Frequency Intervention on a Precision Dart Task

To further validate the causal role of temporal frequencies in precision control, we conduct a controlled zero-out intervention on a dart-to-target task. This experiment complements the main paper's frequency truncation study by exhaustively sweeping chunk sizes and high-frequency removal levels, and observing the resulting execution behavior.

**Experimental setup.** Given an action chunk $a_{t:t+K}$ of length $K$, we compute its spectral representation $c = Fa_{t:t+K}$. We then zero out the highest-frequency coefficients beyond a cutoff index $H$:

$$\hat{c}_i = \begin{cases} c_i, & i \leq K - H, \\ 0, & i > K - H, \end{cases} \qquad \hat{a} = F^{-1}\hat{c}.$$

Here, $K$ denotes the chunk size and $H$ the number of high-frequency components removed. The reconstructed action $\hat{a}$ is replayed in the environment without retraining the policy. This isolates the *execution-level* contribution of high-frequency components from learning effects.

Performance is measured by the final dart landing outcome: *middle hit*, *touching target with error score*, or *toward target but not touching*.

**Results across chunk sizes.** We summarize representative outcomes below, grouped by chunk size $K$.

**Large chunks** ($K = 125$). When only moderate high-frequency components are removed ($H \leq 70$), the dart consistently lands near the center of the target. As more high-frequency components are zeroed out, accuracy degrades smoothly: scores drop to $\sim 9$ for $H \in [80, 100]$, then to $\sim 8$ at $H = 104$–$108$. Beyond $H \geq 110$, the dart no longer touches the target, though the trajectory still points toward it. This indicates that coarse motion remains intact, while fine corrective timing is lost.

**Medium chunks** ($K = 64$). A similar but sharper transition is observed. For $H \leq 32$, the dart lands near the center. As $H$ increases to $40$–$48$, performance degrades gradually (scores $\sim 9$–$8$). For $H \geq 52$, the dart increasingly undershoots and eventually fails to touch the target, despite moving in the correct direction.

**Small chunks** ($K = 32, 16, 8$). With smaller chunks, high-frequency truncation has a more abrupt effect. For $K = 32$, removing nearly all high-frequency components ($H = 31$) creates a large execution gap. For $K = 16$, removing even a single additional high-frequency coefficient ($H = 14 \to 15$) noticeably degrades accuracy. For $K = 8$, zeroing out no high-frequency components yields a centered hit, while removing $5$–$6$ components already causes visible accuracy loss.

**Key observations.** Across all chunk sizes, we observe a consistent pattern:

- Removing high-frequency components does *not* destroy global task intent: the dart continues to move toward the target even under aggressive truncation.

- Precision degrades gradually as more high-frequency content is removed, eventually leading to failure to touch the target.

- Larger chunks tolerate more high-frequency removal before failure, while smaller chunks rely more critically on high-frequency components.

### 7.3. Frequency split selection.

For each task and fixed chunk length $K$, we select the low–high frequency split using a simple, data-driven heuristic based on the spectral statistics of demonstration actions. Given action chunks $a \in \mathbb{R}^{K \times d}$ from $N$ trajectories, we apply the DCT along the temporal dimension and obtain coefficients $c_{k,d}^{(n)}$, $k = 0, \ldots, K-1$, for each demo $n$ and action dimension $d$. We then compute the empirical mean power spectrum

$$\bar{P}_k = \frac{1}{Nd} \sum_{n=1}^{N} \sum_{d=1}^{d} \left| c_{k,d}^{(n)} \right|^2,$$

optionally smoothed with a short moving average, and its cumulative energy

$$E_k = \frac{\sum_{i=0}^{k} \bar{P}_i}{\sum_{i=0}^{K-1} \bar{P}_i}.$$

We first define an energy-based cutoff $K_{\text{energy}}$ as the smallest index satisfying $E_{K_{\text{energy}}} \geq \alpha$ (with $\alpha \approx 0.9$–$0.98$), ensuring that low-frequency components capture most of the spectral energy. In parallel, we detect a "spectral elbow" $K_{\text{elbow}}$ by finding the point of maximum negative curvature in the log-power spectrum $\log(\bar{P}_k + \varepsilon)$ over a restricted range of frequencies, corresponding to the transition from a steep decay (structured motion) to a flatter tail (high-frequency noise and jitter). The final split index is chosen as

$$K_{\text{split}} = \max\left( K_{\text{energy}}, K_{\text{elbow}} \right),$$

optionally snapped to a small set of architecture-friendly values (e.g., $\{4, 8, 16, 24, 32\}$). Coefficients $c_{0:K_{\text{split}}}$ are treated as low-frequency (coarse) components, and $c_{K_{\text{split}}:K}$ as high-frequency (fine) components. Aggregating power across demonstrations and action dimensions makes this procedure substantially more robust than choosing a split from a single trajectory or a single largest amplitude drop.

### 7.4. Extended Method Details for CSP (Causal Spectral Policy)

Here we provide additional details for the CSP (*causal spectral policy*) used in our experiments, including inputs to the policy, the spectral factorization, and model-specific hyperparameters.

For each episode, let $\ell \in \mathbb{R}^{d_\ell}$ be the projected output of a frozen language encoder, $u_t \in \mathbb{R}^{d_u}$ the proprioceptive state (end-effector pose, gripper state), and $o_t$ the visual observation at timestep $t$. All policies—including CSP and baselines—receive the same observation tuple $(o_t, u_t, \ell)$; the baselines concatenate these signals and feed them to a time-domain policy network, while CSP maps them through a perception encoder into a sequence of $N_{\text{obs}}$ tokens.

We assume access to expert actions $a_t \in \mathbb{R}^{D_a}$ (6D twist + 1 gripper) and form length-$T$ action chunks

$$A_i = \{a_i, \ldots, a_{i+T-1}\}.$$

For CSP, the first six action dimensions in each chunk are transformed into the frequency domain using a type-II DCT along time, producing coefficients $C_i \in \mathbb{R}^{T \times 6}$. These coefficients are split into low- and high-frequency subsets $C_i^{\text{low}}$ and $C_i^{\text{high}}$. A GPT-style trunk predicts the low-frequency slice $C_i^{\text{low}}$ from $(o_{i:i+T-1}, u_{i:i+T-1}, \ell)$; a second trunk conditions on the realized low-frequency prediction to infer $C_i^{\text{high}}$. The full coefficient sequence $C_i = [C_i^{\text{low}}, C_i^{\text{high}}]$ is then mapped back to time-domain actions via an inverse

DCT (IDCT), and the gripper channel is concatenated unchanged. CSP is trained with a weighted sum of low- and high-frequency regression losses on these coefficients, as described in Sec. 4.

We use the following CSP-specific hyperparameters in all experiments:

| Parameter | Value |
| --- | --- |
| Action dimension $D_a$ | 7 (6D twist + gripper) |
| Transformer hidden dim $n_{\text{embd}}$ | 256 |
| # Transformer layers | 8 (per stage) |
| # Attention heads | 4 |
| Block size | 65 |
| Predict in frequency | Yes (twist only) |
| Hierarchy | Two-stage (low + high) |

*Table 5.* CSP (causal spectral policy) architecture hyperparameters.

## 7.5. Additional Experimental Data

Fig. 10 provides additional single-task evidence supporting the robustness trends discussed in Sec. 5.3 of the main paper. While the primary experiments focus on aggregated performance across multi-task Libero-90 subsets, this figure disaggregates results at the task level to illustrate how different policy structures respond to realistic teleoperation noise.

Across individual tasks, baseline methods exhibit pronounced task-dependent behavior: some tasks remain relatively stable under noise, whereas others experience sharp performance degradation as noise magnitude increases. This variability is particularly evident for methods that predict actions monolithically in the time domain, where high-frequency execution noise can dominate learning signals.

In contrast, **CSP** consistently shows smaller performance drops across all evaluated tasks and noise regimes. This aligns with the main paper's hypothesis that explicitly modeling action structure across temporal frequencies improves robustness by isolating task-level motion from noisy fine-grained execution. These results further corroborate that the gains reported in Sec. 5 are not driven by a small subset of tasks, but persist across diverse manipulation behaviors with different precision and alignment requirements.

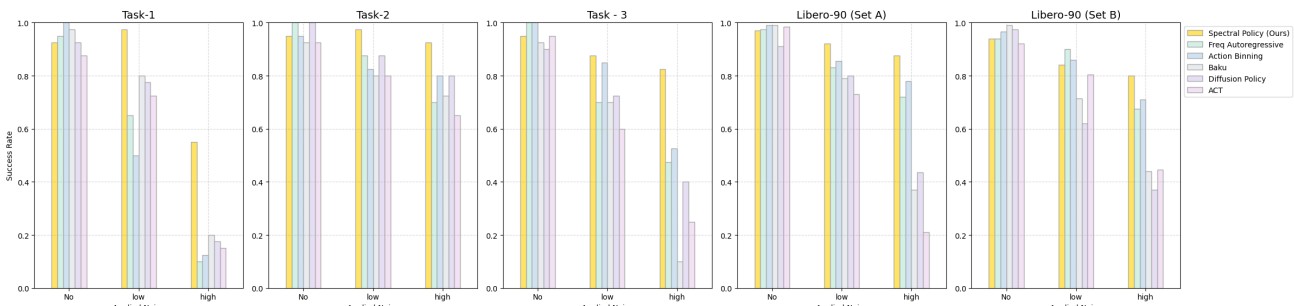

*Figure 10.* **Single-Task Noise Injection Results** Evaluation of **CSP** and baselines on select single tasks from Libero-90.

*Table 6.* **Simulation Results under Additional Action Chunk Sizes.** Success rate across LIBERO and MimicGen-style tasks with a chunk size of K=32.

| Method | Libero-90 | Libero-10 | Stack | Stack3 | Coffee | Square | Threading | MimicGen Mean |
|---|---|---|---|---|---|---|---|---|
| **Chunk Size = 32** | | | | | | | | |
| ACT | 90.3 | 63.9 | 40.0 | 16.3 | 56.3 | 27.5 | 6.3 | 29.3 |
| DP-CNN | 87.0 | 58.1 | 83.8 | 40.0 | 45.0 | 40.0 | 52.5 | 52.3 |
| DP-Transformer | 85.5 | 58.4 | 90.0 | 38.8 | 51.3 | 27.5 | 25.0 | 46.5 |
| BAKU | 82.6 | 49.4 | 67.5 | 17.5 | 32.5 | 26.3 | 7.5 | 30.3 |
| Action Binning | 88.9 | 67.3 | 82.5 | 56.3 | 76.3 | 46.3 | 15.0 | 55.3 |
| Freq-Autoregressive | 88.5 | 68.5 | 85.0 | 32.5 | 67.5 | 27.5 | 28.8 | 48.3 |
| **Spectral Policy (Ours)** | 90.2 | 74.0 | 75.0 | 42.5 | 82.5 | 45.0 | 42.5 | 57.5 |

