# OpenReview forum: "Hierarchical Policy Learning via Spectral Decomposition"
_ICML.cc/2026/Conference — ICML 2026 regular_

### Official Review · Reviewer_h8k6 · 2026-02-13

**Soundness:** 2
**Presentation:** 3
**Significance:** 2
**Originality:** 3
**Overall Recommendation:** 3
**Confidence:** 3

**Summary:**

This paper proposes Causal Spectral Policy (CSP), a hierarchical imitation learning approach that predicts robot actions in the frequency domain via a discrete cosine transform (DCT). CSP factorizes action generation into a coarse-to-fine process: it first predicts low-frequency (coarse trajectory) coefficients from observations and language, then predicts high-frequency (fine corrections) conditioned on the realized coarse trajectory, using stop-gradient to encourage a unidirectional dependence. The paper presents frequency truncation diagnostics, simulation benchmarks (LIBERO and MimicGen-style tasks), structured noise-injection experiments modeling teleoperation artifacts, and real-robot evaluations on precision-critical tasks. Overall, the work offers an intuitive spectral perspective on hierarchical action generation and provides promising empirical evidence, but the current evaluation does not fully establish whether improvements are driven by spectral decomposition itself or primarily by the two-stage hierarchical structure and additional computation/capacity, due to missing attribution-focused controls.

**Compliance With Llm Reviewing Policy:**

Affirmed.

**Key Questions For Authors:**

1. CSP uses a two-stage architecture (two trunks / two passes). Did you run a parameter- and compute-matched two-stage time-domain baseline that (i) predicts a coarse trajectory first and (ii) predicts a conditional refinement given the realized coarse trajectory, without using DCT? If yes, please report results; if not, please explain why and provide an estimate.

2. Can you clarify the intended factorization? Specifically, the paper seems to generate high-frequency coefficients conditioned on the observation and the realized low-frequency trajectory, rather than conditioning high-frequency coefficients on observation and language alone. What assumptions justify conditioning on the realized low-frequency trajectory, and how does stop-gradient relate to this choice? Beyond coefficient truncation, could you add (or describe) an additional interventional analysis that would better support the causal interpretation?

3. How are chunk boundaries handled at inference time (stride/overlap-add/warm starts)? Do you observe boundary artifacts, and if so, how are they mitigated?

4. DCT is a global basis within a chunk and may poorly localize impulsive/contact-rich events. Do you have evidence (qualitative or quantitative) on failure modes where DCT localization is problematic, or a lightweight comparison to a more localized basis (e.g., wavelets) / learned dictionary?

**Limitations:**

Yes

**Strengths And Weaknesses:**

Strengths:

Clear and intuitive formulation. Decomposing actions into low-frequency intent and high-frequency refinements, implemented as a two-stage predictor, is simple, easy to follow, and aligns well with common “plan-then-refine” intuition in manipulation.

Informative diagnostics. The frequency truncation / replay study is a useful diagnostic that helps connect high-frequency content to precision and timing behaviors, lending support to the design choice.

Practical robustness testbed. The proposed human-inspired teleoperation noise model (temporally correlated drift with intermittent corrective bursts) is well-motivated and could be broadly useful for evaluating robustness to imperfect demonstrations.

Reasonable evaluation breadth. Results are reported across clean simulation, structured noise robustness tests, and real-robot tasks emphasizing alignment/contact, with generally positive trends for CSP in precision-sensitive settings.

Weaknesses

Key attribution control is missing. CSP uses a two-stage architecture (effectively two trunks / two passes). Without a capacity- and compute-matched two-stage time-domain baseline—e.g., first generating a coarse trajectory and then predicting a conditional refinement in the time domain—it is difficult to rule out that gains stem mainly from hierarchical refinement and extra computation rather than the frequency-domain (DCT) choice.

“Causal” framing may be overstated. The claimed causality is largely architectural (conditioning + stop-gradient) rather than causal identification in an SCM sense. A clearer causal graph/assumptions, and additional interventional evidence beyond coefficient truncation, would strengthen the claim.

Representation assumptions may limit generality. DCT is a chunk-level global basis and implicitly favors within-chunk stationarity/smoothness; contact-rich or impulsive events may be poorly localized. The paper does not explore (or even lightly compare against) alternatives with better time-frequency localization (e.g., wavelets or learned dictionaries), leaving open questions about when the spectral choice is truly beneficial.

Action-dimension decoupling. Applying per-dimension DCT ignores cross-DoF coupling, which may limit expressivity for behaviors requiring tightly coordinated multi-DoF changes.

---

> ### Author Rebuttal · Authors · 2026-03-31
>
> We thank Reviewer for the detailed and insightful feedback. We address the concerns below. **More detail in webpage**
>
> **Weakness 1 & Question 1: Attribution (spectral vs hierarchy)**
>
>
> We agree that isolating the source of improvement is important. Our key contribution is the **coarse-to-fine (C2F) decomposition in the spectral domain, not hierarchy alone**.
> All baselines and ablations use comparable or larger model sizes, ruling out parameter effects. Empirically, frequency-only models (no hierarchy) underperform CSP (Table 1). In the time domain, **Action Binning** serves as a hierarchical baseline, first predicting a coarse discretized action bin and then refining it with continuous residual actions, forming a two-stage coarse-to-fine structure; however, despite this hierarchy, its performance remains below CSP, particularly at longer horizons (Table 2). See webpage for detailed structure.
> This shows hierarchy alone is insufficient, and spectral parameterization is critical for separating global motion from fine corrections. While a fully compute-matched two-stage time-domain baseline would strengthen attribution, current results already indicate both are necessary
>
>
> **Weakness 2 & Question 2: “Causal” framing and factorization**
>
>
> We clarify that “causal” refers to a structural factorization, not causal identification. We assume coarse actions depend on observation and language, while fine actions depend on the realized coarse trajectory. The stop-gradient enforces this direction by preventing fine losses from altering the coarse prediction.
> We validate this via counterfactual interventions (holding observation fixed and modifying conditioning):
>
>
> | Low-Freq Condition         | High-Freq Condition                          | L90 (K=16) | L90 (K=32) | L10 (K=16) | L10 (K=32) |
> |----------------------------|-----------------------------------------------|------------|------------|------------|------------|
> | obs                        | language + obs + low-freq                     | 22.22      | 18.22      | 64.5       | 61         |
> | language + obs             | low-freq                                      | 74.33      | 69.89      | 68.5       | 66.5       |
> | language + obs             | obs                                           | 86.22      | 84.22      | 65         | 51.5       |
> | language + obs             | low-freq + language                           | 91.56      | 90.44      | 81         | 75         |
> | high-freq + obs            | language + obs       | 49.56      | 46         | 35.5       | 55.5       |
> | high-freq + language + obs | language + obs       | 80.33      | 66.78      | 49.5       | 46         |
>
>
>
> Removing language in the coarse stage causes a large drop (91.56 → 22.22), showing coarse depends on observation + language. Removing coarse conditioning in the fine stage degrades performance (→ 86.22 / 65), showing fine depends on the realized coarse trajectory. Reversing the dependency further drops performance (→ 49.56), showing the dependency is directional.
> These trends are consistent across chunk sizes, demonstrating that fine predictions depend on the realized coarse motion and supporting the proposed factorization.
>
> **Weakness 3 & Questions 3–4: Representation and inference**
>
>
> The DCT provides an orthonormal temporal decomposition that separates motion into frequency scales with minimal complexity. Empirically, low-frequency components capture global motion structure, while high-frequency components capture contact timing and alignment corrections, which are handled by the refinement stage.
> We use receding-horizon inference (predict a chunk and execute the first step), avoiding boundary artifacts and ensuring smooth transitions; we do not observe discontinuities. For fair comparison, we fix the action horizon = 8 across all methods and evaluate chunk sizes 16, 32, 64. Horizon 8 is commonly used in chunk-based behavior cloning and balances temporal context with training stability.
> While more localized bases (e.g., wavelets) improve temporal localization, they lose global structure. In contrast, the global DCT basis captures long-range dependencies, which supports modeling the coarse-to-fine relationship between overall motion intent and local corrections, reflecting a tradeoff between localization and global structure.
>
>
>
> **Weakness 4 & Question 5: Action-dimension decoupling**
>
>
> Although DCT is applied per action dimension, this does not limit expressivity. DCT is an orthonormal reparameterization along the temporal axis, while cross-DoF coordination is still modeled jointly by the shared policy network, preserving coupled behaviors across joints.
> We also experimented with applying DCT jointly across all action dimensions, including the gripper, at an early stage. However, the gripper signal typically contains sparse or binary changes and does not exhibit strong spectral structure, which degraded performance when transformed together with continuous motion dimensions.

---

> > ### Author Rebuttal · Reviewer_h8k6 · 2026-04-01
> >
> > Thank you for the rebuttal and the additional clarifications. The response is helpful, especially in clarifying the intended structural factorization, the role of stop-gradient, and the rationale behind the spectral decomposition. However, my main concern is only partially addressed. In particular, the key attribution issue remains unresolved: without a parameter- and compute-matched two-stage time-domain baseline that first predicts a coarse trajectory and then refines it conditionally, it is still difficult to determine how much of the gain comes from spectral decomposition itself rather than from the hierarchical two-stage design and additional computation. In addition, the “causal” interpretation is now more carefully framed as a structural factorization, but this still does not fully justify the original causal terminology. The concerns regarding the limitations of DCT as a global basis and the lack of comparison to more localized alternatives also remain. Therefore, while the rebuttal improves the paper, my overall evaluation remains unchanged.

---

> > > ### Author Response · Authors · 2026-04-07
> > >
> > > We thank the reviewer for the clarification.
> > >
> > >
> > > **Two-stage Time-domain Baseline**
> > >
> > > Regarding the attribution concern, we believe the requested parameter- and compute-matched two-stage time-domain baseline is already included in our paper as **Action Binning**, which fulfills the requirement of a hierarchical coarse-to-fine design.
> > >
> > > The action binning formulation is:
> > >
> > > Let $a$ denote one scalar action value and let $n$ be the stride.
> > >
> > > The coarse stage predicts the large-scale grid location:
> > > $$
> > > \{0, n, 2n, 3n, \dots\}.
> > > $$
> > >
> > > The coarse label is
> > > $$
> > > y^{(c)} = \left\lfloor \frac{a}{n} \right\rfloor .
> > > $$
> > >
> > > The fine stage predicts the local offset within each coarse block:
> > > $$
> > > \{0,1,2,\dots,n-1\}.
> > > $$
> > >
> > > Together, the resulting action values cover the full discrete space:
> > > $$
> > > \{0,\dots,n-1\},\;
> > > \{n,\dots,2n-1\},\;
> > > \{2n,\dots,3n-1\},\;
> > > \dots
> > > $$
> > >
> > > The fine label is
> > > $$
> > > y^{(f)} = a - n y^{(c)} .
> > > $$
> > >
> > > Thus the action is decomposed as
> > > $$
> > > a = n y^{(c)} + y^{(f)} .
> > > $$
> > >
> > > The prediction factorization is
> > > $$
> > > p(a|o,c) = p_c(y^{(c)}|o,c)\; p_f(y^{(f)}|o,c,y^{(c)}).
> > > $$
> > >
> > > We ensure that the **architecture, parameter scale, and training setup remain comparable** between CSP and Action Binning.
> > > Empirically, Action Binning consistently underperforms CSP, especially for longer chunks, indicating that the improvement does not arise from hierarchy alone, but from the spectral parameterization, which better separates global motion structure from fine corrective behavior.
> > >
> > >
> > >  **Clarification of term causal**
> > >
> > >
> > > Regarding the use of the term “causal”, we understand the reviewer’s concern. Our use of causal refers to a structured conditional dependency, not causal discovery. Specifically, low-frequency components depend on observation and language, while high-frequency components depend on the realized low-frequency trajectory and observation, rather than hierarchy alone. We will clarify this terminology more precisely in the revision.
> > >
> > >
> > > **Localized alternative spectral bases**
> > >
> > > Our goal is not to advocate a particular transform, but to analyze the **semantic role of temporal frequency** in robot actions. We consistently observe a **coarse-to-fine structure: low frequencies encode global motion intent**, while **high frequencies encode local corrective behavior**. The transform mainly serves to reveal this structure.
> > > We adopt **DCT** because it provides a **global orthonormal basis with a fixed frequency ordering**, yielding a **stable and task-consistent coarse-to-fine decomposition** across chunk sizes. This global structure aligns with the observation that long-horizon manipulation trajectories exhibit smooth global structure with localized corrective adjustments. In contrast, **wavelets emphasize temporal locality, but introduce additional design choices (mother wavelet, scale depth, boundary handling) and the optimal frequency split varies across tasks, requiring extra tuning. Importantly, wavelets do not provide a consistent ordering that reliably separates global planning from fine correction across manipulation settings**.
> > >
> > > We evaluate on LIBERO-10, MimicGen square_d1, and stack_d1, selecting the best wavelet split per chunk size (WThis experiment required significant tuning effort compared with CSP):
> > > Wavelet vs DCT (best wavelet split per chunk size)
> > >
> > > Task          | Chunk | Wavelet (best) | CSP (DCT)
> > > --------------|-------|----------------|-----------
> > > LIBERO-10     | 16    | 0.750          | **0.808**
> > > LIBERO-10     | 32    | **0.745**          | **0.745**
> > > square_d1     | 16    | 0.317          | **0.488**
> > > stack_d1      | 16    | 0.883          | **0.925**
> > > stack_d1      | 32    | **0.800**          | 0.75
> > >
> > >
> > > Wavelets require additional design choices (e.g., split level), and the optimal partition varies across tasks, leading to substantial tuning effort. Results further confirm that **hierarchical modeling in the spectral domain improves over naïve action-space learning**, as both wavelet and DCT decompositions outperform non-spectral baselines. However, despite additional tuning, wavelets do not consistently outperform DCT. In contrast, **DCT provides a fixed frequency ordering and stable coarse-to-fine structure** across chunk sizes and tasks with minimal tuning. This indicates that the **global spectral basis of DCT is already sufficient and more robust**, capturing the hierarchical structure without introducing extra hyperparameters.

---

### Official Review · Reviewer_GfZA · 2026-02-24

**Soundness:** 3
**Presentation:** 3
**Significance:** 3
**Originality:** 3
**Overall Recommendation:** 4
**Confidence:** 3

**Summary:**

This paper delves into a core contradiction in robot imitation learning: the conflict between long-horizon task planning and high-precision contact manipulation. The authors observe that current action chunking strategies treat all timesteps equally in the time domain, which causes models to struggle with maintaining high-frequency details under long prediction windows. To address this, the paper proposes the Causal Spectral Policy (CSP), which maps action sequences to the frequency domain using the Discrete Cosine Transform (DCT) and introduces a coarse-to-fine causal generation architecture. It first predicts the low-frequency global trajectory based on language and visual observations, and then predicts high-frequency fine-grained corrective actions conditioned on the low-frequency trajectory. This method not only demonstrates excellent resistance to degradation in long action sequence prediction but also introduces a noise injection model that aligns with human teleoperation habits, proving CSP's strong robustness on imperfect demonstration data containing high-frequency noise.

**Compliance With Llm Reviewing Policy:**

Affirmed.

**Final Justification:**

Thanks for the detailed rebuttal by the authors, now i have no concerns.

**Key Questions For Authors:**

1. Compared to standard single-forward-pass baselines, what is the exact average inference latency of the cascaded CSP architecture in your real-robot experiments? It would be helpful to clarify if this cascaded prediction causes any observable control stuttering or jitter during deployment.
2. Since gripper actions are concatenated unchanged outside the frequency domain, how does the model ensure strict frame-level synchronization between the discrete gripper closing events and the continuous spatial corrective actions predicted by the high-frequency components?
3. The provided anonymous link is currently empty. Could you please provide comprehensive real-robot execution videos during the rebuttal phase? Specifically, side-by-side comparisons between CSP and baselines (like ACT) during the high-precision contact phases would be instrumental in visually proving the superior high-frequency control capabilities of your method.

**Limitations:**

yes

**Strengths And Weaknesses:**

# Strength
1. The paper validates its approach through an insightful physical dart insertion experiment, proving that low frequencies dictate global intent while high frequencies govern contact and alignment. The resulting two-stage causal graph $P_{low}$ and $P_{high}$ closely matches human motor control intuitions.
2. The authors introduce a highly valuable teleoperation noise model featuring temporally correlated drift and bursty high-frequency corrections. CSP demonstrates impressive robustness against this imperfect, noisy demonstration data.
3. The paper presents a robust long-sequence prediction. Standard chunking methods suffer from over-smoothing at long horizons. In experiments with a highly challenging chunk size of $K=64$, baseline methods like ACT and BAKU drop below a 20% success rate, whereas the proposed CSP maintains a 50% success rate.

# Weakness
1. Spectral decomposition techniques like DCT excel on smooth continuous trajectories but struggle with discrete signals like gripper actions. The appendix reveals that the gripper channel is merely concatenated unchanged, which compromises the mathematical elegance of a unified spectral action space and may disrupt the synergy between gripper timing and high-frequency arm corrections.
2. CSP relies on a cascaded architecture involving two sequential Transformer backbones. The manuscript lacks discussion on wall-clock inference latency or control frequency (Hz) during real-world deployment. Sacrificing control frequency for high-frequency prediction accuracy would be a significant practical trade-off.
3. Although the paper reports impressive real-robot results in Figure 6 and Table 3 and provides an anonymous project page link, the linked webpage is currently empty. Given that the core contribution is improving high-frequency execution (e.g., reducing jitter, improving contact alignment), static keyframe images are insufficient to evaluate motion smoothness and corrective behaviors. The lack of video evidence severely weakens the credibility of the real-world deployment claims.

---

> ### Author Rebuttal · Authors · 2026-03-31
>
> We thank the reviewer for the positive assessment and insightful suggestions. We address the concerns below.
>
>
> **Weakness 1 & Question 2: Gripper modeling and synchronization**
>
>
>
> We intentionally do not apply DCT to the gripper channel, as it is **discrete, event-driven**, and frequency decomposition is not meaningful for binary signals. CSP therefore adopts a hybrid representation: continuous arm actions are modeled in the spectral domain, while gripper actions remain in the time domain and are predicted jointly. This does not break mathematical consistency, since DCT is an **orthonormal per-dimension reparameterization** applied only to continuous channels. The gripper is treated as an additional dimension outside this transform and does not interfere with spectral decomposition.
>
>
>
> Cross-dimension coordination is captured by the shared policy network, which jointly predicts all outputs, allowing it to learn correlations between high-frequency motion corrections and gripper events. Since gripper actions do not exhibit a coarse-to-fine structure, we treat them as high-frequency decisions aligned with the refinement stage. In practice, the model learns consistent co-occurrence between corrections and gripper events (e.g., contact), resulting in accurate **frame-level synchronization**.
>
>
>
> We also experimented with applying DCT jointly across all action dimensions, including the gripper, at an early stage. However, the gripper signal typically contains sparse or binary changes and does not exhibit strong spectral structure, which degraded performance when transformed together with continuous motion dimensions.
>
>
>
> **Weakness 2 & Question 1: Inference latency and control stability**
>
>
>
> CSP does not reduce control frequency or introduce jitter. In our real-robot setup, all methods operate within the standard control range (10–100 Hz) used across benchmarks, with CSP running at the same frequency as baselines.
>
>
>
> CSP performs two lightweight forward passes within a single control step, both executed within the same control cycle, resulting in latency comparable to standard Transformer-based policies (e.g., ACT/BAKU), without affecting real-time execution.
>
>
>
> The policy runs at 10 Hz on the real robot. We observe **no stuttering or delay** in deployment, as both coarse and fine components are predicted within the same timestep. (See webpage for the video) Increasing control frequency mainly changes temporal discretization, while the spectral structure remains stable, so CSP does not require higher control rates or incur additional latency overhead.
>
>
>
> **Weakness 3 & Question 3: Real-robot evidence**
>
>
> We apologize for the incomplete webpage. We have updated the webpage to include:
>
>
> **• Full real-robot execution videos**
>
>
> **• Side-by-side comparisons with baselines**
>
>
> **• Videos under simulated teleoperation noise**
>
>
> **• Additional ablation results**
>
>
>
> These will clearly demonstrate improved alignment, reduced jitter, and more stable contact behavior, which are difficult to convey via static images.

---

> > ### Author Rebuttal · Reviewer_GfZA · 2026-04-02
> >
> > Thanks for the detailed rebuttal by the authors, now i have no concerns.

---

> > > ### Author Response · Authors · 2026-04-07
> > >
> > > We sincerely thank the reviewer for the time and effort spent reviewing our paper and for the thoughtful and constructive feedback. We are encouraged by the positive assessment and greatly appreciate the valuable suggestions, which have helped improve the clarity and quality of the work.

---

### Official Review · Reviewer_vyfN · 2026-03-13

**Soundness:** 3
**Presentation:** 4
**Significance:** 3
**Originality:** 3
**Overall Recommendation:** 4
**Confidence:** 2

**Summary:**

This paper proposes Causal Spectral Policy (CSP) for robot imitation learning by decomposing action chunks in the DCT spectral domain into low-frequency coarse motion and high-frequency fine corrections, and modeling them with an explicit causal coarse-to-fine factorization. The paper argues, through frequency truncation analysis and real-robot rollouts, that low-frequency components preserve global motion intent while high-frequency components are critical for precise alignment and contact. Empirically, CSP is evaluated on LIBERO, MimicGen-style tasks, simulated teleoperation-noise settings, and several real-robot manipulation tasks, where it generally outperforms time-domain and frequency-domain baselines, especially on precision-sensitive and noisy settings.

**Compliance With Llm Reviewing Policy:**

Affirmed.

**Final Justification:**

The explanations were helpful for understanding the paper. I am keeping the score.

**Key Questions For Authors:**

- In your intervention experiments (Section 3.2), did you observe any tasks where low-frequency components depended on high-frequency feedback, or is the coarse-to-fine direction strictly unidirectional in all tested manipulation scenarios?

- You mentioned a data-driven heuristic for the $\lambda$ cutoff (80% energy). Does this threshold need to be re-tuned for robots with significantly different control frequencies (e.g., 20Hz vs 500Hz)?

- Figure 6 shows failure cases like "misaligned stacking." Does the high-frequency module typically fail due to perceptual errors (observation noise) or an inability to recover from a poor initial coarse trajectory?

- Can you provide more quantitative evidence (e.g., power spectrum comparison) showing that your injected "TeleOp Noise" matches the distribution of actual human jitter measured during real teleoperation?

**Limitations:**

yes

**Strengths And Weaknesses:**

Strengths:

- The decomposition of robot actions into task-level intent (low-freq) and execution-level refinements (high-freq) is well-motivated and supported by spectral analysis.

- The proposed CSP framework is clean, easy to implement, and provides a principled way to handle different temporal scales.

- CSP shows significant advantages in precision-sensitive tasks and maintains robustness even as action chunk sizes increase, where traditional methods like ACT or Diffusion Policy often degrade.

- The introduction of human-inspired teleoperation noise injection as a data augmentation technique is a practical contribution to real-world imitation learning.

Weaknesses:

- While the paper uses "Frequency-Domain Intervention" to motivate the architecture, the theoretical link between this spectral truncation and formal causal graphical models could be more explicitly defined to justify the "Causal" label in CSP.

- Although the real-robot experiments cover diverse tasks (typing, stacking, assembly), the evaluation is conducted in relatively static environments. It remains unclear how CSP handles dynamic scene changes during the "fine-correction" phase.

- The hierarchical nature requires two separate prediction modules. A more detailed discussion on the training wall-clock time and inference latency compared to single-stage diffusion policies is missing.

- While imitation learning baselines are strong, comparing against recent hierarchical representations (e.g., those using VQ-VAEs for action primitives) would further strengthen the claim that spectral decomposition is superior to latent decomposition.

---

> ### Author Rebuttal · Authors · 2026-03-31
>
> We thank Reviewer for the positive assessment and insightful questions. We are encouraged that the coarse-to-fine spectral decomposition and robustness results are found meaningful. We address the concerns below. **More detail in webpage**
>
>
> **Weakness 1 & Question 1: “Causal” interpretation and directionality**
>
>
> We clarify that “causal” refers to a **structural coarse-to-fine dependency**, not formal causal identification. Our formulation assumes low-frequency components encode global motion intent, while high-frequency components provide local refinement conditioned on this structure.
>
>
> We validate this via **asymmetric intervention and counterfactual experiments**, summarized below:
>
>
> | Low-Freq Condition         | High-Freq Condition                          | Libero90 (K=16) | Libero90 (K=32) | Libero10 (K=16) | Libero10 (K=32) |
> |----------------------------|-----------------------------------------------|------------|------------|------------|------------|
> | obs                        | language + obs + low-freq                     | 22.22      | 18.22      | 64.5       | 61         |
> | language + obs             | low-freq                                      | 74.33      | 69.89      | 68.5       | 66.5       |
> | language + obs             | obs                                           | 86.22      | 84.22      | 65         | 51.5       |
> | language + obs             | low-freq + language                           | 91.56      | 90.44      | 81         | 75         |
> | high-freq + obs            | language + obs (predict high-freq first)      | 49.56      | 46         | 35.5       | 55.5       |
> | high-freq + language + obs | language + obs (predict high-freq first)      | 80.33      | 66.78      | 49.5       | 46         |
>
>
> Each row corresponds to a **targeted intervention on the dependency structure**. When the low-frequency stage excludes language (first row), performance drops sharply (91.56 → 22.22), showing that coarse motion depends on observation and language. When the high-frequency stage does not take low-frequency input (third row), performance degrades (→ 86.22 / 65), indicating that fine corrections depend on the realized coarse trajectory. Critically, reversing the dependency by predicting high-frequency first (last two rows) leads to a substantial drop (→ 49.56), demonstrating that the dependency is **directional rather than symmetric**. These trends are consistent across chunk sizes.
>
>
> In addition, frequency truncation experiments show that removing low-frequency components leads to small-amplitude jitter without meaningful motion, while removing high-frequency components preserves the overall trajectory but reduces precision near contact.
>
>
>
> **Weakness 2 & Question 3: Failure modes and robustness**
>
>
> Misaligned stacking illustrates a typical failure mode of **baseline methods**. Compared to these baselines, **CSP significantly reduces such errors**, indicating stronger capability in fine-grained motion correction. When failures still occur (e.g., misaligned stacking), they are primarily caused by an incorrect coarse trajectory, rather than limitations of the fine stage.
>
>
>
> **Weakness 3 & Question 2: Frequency cutoff and control frequency**
>
>
> We evaluate control frequencies from 10–100 Hz and observe consistent spectral energy distributions. Increasing control frequency mainly increases temporal resolution, while the relative frequency structure remains stable, so the 80% energy cutoff does not require retuning.
>
>
> We further perform a frequency selection analysis within a fixed chunk size (K = 32); results (see new line chart on the webpage) show performance is stable across a broad range of splits, with an optimum around mid frequencies. Across chunk sizes {16, 32, 64} and multiple tasks, the best performance consistently falls within a similar split range, indicating the 80% energy heuristic generalizes well across manipulation settings
>
>
> **Weakness 4 & Question 4: Teleoperation noise realism**
>
>
> We validate realism by comparing injected noise with real teleoperation trajectories using distributional metrics and qualitative analysis (see webpage). We collect 20 teleoperation demos per task using keyboard + SpaceMouse, and compute KL divergence in both action space and frequency space across chunk sizes {16, 32, 64}. Synthetic noise consistently produces lower KL divergence to teleoperation data:
>
>
> Orig vs Teleop: Action KL 9.5–11.7, Freq KL 0.50–1.06
>
>
>  Syn vs Teleop: Action KL 7.1–8.4, Freq KL 0.42–0.82
>
>
> This indicates improved alignment with human motion variability, particularly temporally correlated jitter and bursty high-frequency corrections. Metrics and rollout videos are provided on the webpage.

---

> > ### Author Rebuttal · Reviewer_vyfN · 2026-04-04
> >
> > Thank you for the rebuttal. The explanations were helpful for understanding the paper.

---

> > > ### Author Response · Authors · 2026-04-07
> > >
> > > We sincerely thank the reviewer for the time and effort spent reviewing our paper and for the constructive and thoughtful feedback throughout the process. We are greatly encouraged that the rebuttal has addressed the concerns, and we truly appreciate the valuable suggestions that helped improve the clarity and quality of the work.

---

### Official Review · Reviewer_DCsM · 2026-03-17

**Soundness:** 3
**Presentation:** 3
**Significance:** 2
**Originality:** 2
**Overall Recommendation:** 4
**Confidence:** 4

**Summary:**

The paper introduces causal spectral policy, a coarse-to-fine imitation learning architecture. The key idea is to do DCT on action chunks, and predict low-frequency components first and then high-frequency components conditioned on the low-freq prediction. Experiments in sim and real shows that CSP outperforms baseline behavior cloning methods.

**Compliance With Llm Reviewing Policy:**

Affirmed.

**Final Justification:**

See rebuttal acknowledgement.

**Key Questions For Authors:**

- In Figure 5, can we show the performance curve across the full spectrum of K to probe when it breaks?
- Clarification: for the experiments in Table 1, could you clarify whether all baselines predict action chunks of 16 and 64, and whether it's rolled out open loop within each chunk, or with receding horizon control?
- Could you elaborate on how using a larger chunk size for manipulation improves performance? It seems from Table 1 that the larger chunk size overall performs worse across policy classes.
- The real results in Table 3 for diffusion policy are surprisingly poor. Could you elaborate on the failure modes and provide details on hyperparameter / implementation?
- Clarification: what is the difference between "frequency autoregressive", "frequency diffusion", and "no hierarchy" in Section 5.1.2?

**Limitations:**

- The method uses a static cutoff of frequency components into low and high parts.
- The evaluations for noise robustness uses synthetic noise.

**Strengths And Weaknesses:**

- The paper is clearly written and easy to follow, and the idea of coarse-to-fine action modeling in the spectral domain is well-motivated.
- The empirical validation is comprehensive with simulated and real robot results, showing superior performance over state-of-the-art behavior cloning methods.
- Limited novelty and unclear significance. DCT on action chunks have become standard. The core claim is that this helps with performance at larger chunk sizes and is robust to synthetic human-like noise. The paper argues that large chunks helps with consistency but it does not clearly demonstrate this in downstream evaluations. And the noise robustness evaluation is synthetic and it seems unclear whether this corresponds to failure modes with policies trained on real datasets.

---

> ### Author Rebuttal · Authors · 2026-03-31
>
> We thank Reviewer for the thoughtful and constructive suggestions. We address the concerns below. **More detail in webpage**
>
> **Weakness (Novelty) & Questions 1–3: Role of DCT, chunk size, and frequency selection**
>
>
> We agree that **DCT-based representations** have appeared in recent work, partly influenced by **Fast Tokenization**. However, our contribution is fundamentally different. Prior works use DCT mainly as a **compact parameterization** for sequence compression, where higher **control frequency** leads to more steps within a fixed time window. In contrast, our work studies the **semantic role of temporal frequency**, where low-frequency components capture coarse motion structure and high-frequency components capture fine corrective behavior.
> This interpretation is tightly coupled with **chunk size (Q3**). Since DCT is applied along the **temporal dimension**, the meaning of each frequency component depends on the number of steps per action chunk. Thus, chunk size is a core modeling factor, not merely an implementation detail.
> To address **Q1**, we perform a **frequency selection analysis** (complete line chart in webpage) within a fixed chunk size (**K = 32**), varying how many **low-frequency components** are retained:
> | # Low-Frequency Components | 2 | 4 | 6 | 8 | 10 | 12 | 14 | 16 | 18 | 20 | 22 | 24 | 26 | 28 | 30 |
> | -------------------------- | ---- | ----- | ----- | ----- | ----- | ----- | ----- | ----- | ----- | ----- | ----- | ----- | ----- | ----- | ----- |
> | Success (Libero-90) | 0.89 | 0.889 | 0.908 | 0.908 | 0.919 | 0.917 | 0.906 | 0.901 | 0.899 | 0.888 | 0.868 | 0.881 | 0.858 | 0.854 | 0.857 |
>
> Performance is **stable across a wide range of splits**, with a broad optimum near mid frequencies, indicating the method is not sensitive to the cutoff and supporting our energy-based (~80%) selection. Across chunk sizes {16, 32, 64}, **CSP remains robust** while baselines degrade more at larger chunks.
> Together, these results clarify: chunk size defines frequency semantics,  frequency split defines coarse-to-fine decomposition, CSP is robust to both
>
>
> This explains why larger chunks help (Q3): they provide longer temporal context, while CSP preserves precision via explicit high-frequency refinement.
>
>
>
> **Weakness & Question 2: Experimental setup (chunk vs horizon)**
>
>
> We fix action horizon = 8 across all methods for fair comparison. Chunk size controls temporal grouping, while horizon controls rollout length, following standard practice for consistent evaluation.
>
>
>
> **Weakness & Question 4: Real-world performance and diffusion policy behavior**
>
>
> The diffusion baseline appears weak due to **strict success criteria**. Keyboard tasks require pressing **only the target key** without contacting neighbors (see videos). Under a relaxed criterion, performance improves (e.g., **9/10 Enter, 7/10 C**). For stacking, diffusion achieves **6/10** under **precision-sensitive evaluation**.
>
>
> These observations motivated analysis of high-frequency corrections, as demonstrations often contain **rapid adjustments near contact**. Prior theory shows diffusion noise schedules attenuate high-frequency components more strongly, making fine corrections harder to recover (Falck et al., 2025, A Fourier Space Perspective on Diffusion Models), explaining the behavior in precision-sensitive tasks.
>
>
>
> **Weakness & Question 5: Ablation clarity**
>
>
> All ablations use the **same architecture** and **parameter count**, differing only in **frequency modeling (see webpage)**.
> Frequency Autoregressive: sequential spectral prediction without hierarchy
>
>
>  Frequency Diffusion: joint spectral prediction without conditioning
>
>
>  No Hierarchy: single-stage prediction
>
>
> These isolate the effect of hierarchical spectral factorization, independent of model capacity.
>
>
>
> **Limitations**
>
>
> CSP currently uses a fixed frequency cutoff. While performance is stable across a wide range of splits, the optimal boundary may be task-dependent, suggesting future work on adaptive frequency partitioning.
> We agree robustness evaluation uses synthetic teleoperation noise. To validate realism, we collect 20 teleoperation demonstrations per task using keyboard + SpaceMouse and compare distributions via KL divergence in action space and frequency space across chunk sizes {16, 32, 64}:
>
>
> | Metric | Action KL | Freq KL (16) | Freq KL (32) | Freq KL (64) |
> | -------------- | ----------- | ------------- | ------------- | ------------- |
> | Orig vs Teleop | 9.5–11.7 | 0.50–0.71 | 0.65–0.74 | 0.85–1.06 |
> | Syn vs Teleop | **7.1–8.4** | **0.42–0.55** | **0.52–0.60** | **0.69–0.82** |
>
>
>
>
> Synthetic noise produces trajectories closer to real teleoperation data, indicating better alignment with human motion variability, especially in temporal frequency characteristics. While large-scale teleoperation collection is costly, the KL analysis suggests the synthetic noise captures key statistical properties of human corrective behavior relevant for robust policy learning.

---

> > ### Author Rebuttal · Reviewer_DCsM · 2026-04-06
> >
> > Thank you for the rebuttal response. I have decided to increase my score given the additional experiments and clarifications. One remaining concern is that using much longer action chunks seem to degrade performance from the experiments, so it is unclear why we would want to use very long action chunks in the first place. If the core claim is temporal consistency / performance benefits, it would be good to support it with a set of experiments explicitly testing whether longer chunks outperform shorter chunks in relevant settings.

---

> > > ### Author Response · Authors · 2026-04-07
> > >
> > > We sincerely thank the reviewer for the time and thoughtful feedback, and we are encouraged that the additional clarifications were helpful.
> > >
> > > Because spectral representations are inherently defined over action chunks, we vary chunk sizes to verify that the proposed coarse-to-fine frequency structure remains consistent across temporal resolutions. Our goal is not to claim longer chunks are always better, but to show that CSP remains stable as chunk size changes, whereas baselines degrade more significantly, indicating the improvement comes from the spectral decomposition rather than a specific chunk length choice

---

### Decision · Program_Chairs · 2026-04-30

**Decision:**

Accept (regular)

**Comment:**

This paper proposes Causal Spectral Policy (CSP) a course-to-fine action model based on spectral decomposition of action trajectories in which high-level coarse actions are refined by finer corrections to realize precise action trajectories.  The paper is well-written and the method is principled.  The evaluation is extensive with validation in real and sim against strong baselines including those which show the value of spectral decomposition.  The experiments show the method is more robust to action chunk length.  The human teleop inspired noise augmentation was widely praised by reviewers.  Several reviewers noted the paper could benefit from more analysis of the time costs of the method.  Although the authors addressed this at a high level, noting the model runs at 10hz in real, I believe more detail here would be beneficial. Another limitation pointed out by reviewers was the fact that methods which use DCT are not as effective for intermittent actions such as grippers closing or responding to sudden impulsive or contact-rich events. The gripper action, for example, is dealt with by leaving this out of the spectral action space, suggesting the approach may not be beneficial across the board and requiring care in its use.